# Expanding the Bacterial Diversity of the Female Urinary Microbiome: Description of Eight New *Corynebacterium* Species

**DOI:** 10.3390/microorganisms11020388

**Published:** 2023-02-03

**Authors:** Elisabete Alves Cappelli, Magdalena Ksiezarek, Jacqueline Wolf, Meina Neumann-Schaal, Teresa Gonçalves Ribeiro, Luísa Peixe

**Affiliations:** 1Associate Laboratory i4HB—Institute for Health and Bioeconomy, Faculty of Pharmacy, University of Porto, 4050-313 Porto, Portugal; 2UCIBIO—Applied Molecular Biosciences Unit, Department of Biological Sciences, Faculty of Pharmacy, University of Porto, 4050-313 Porto, Portugal; 3Leibniz Institute DSMZ—German Collection of Microorganisms and Cell Cultures GmbH, 38124 Braunschweig, Germany; 4CCP—Culture Collection of Porto, Faculty of Pharmacy, University of Porto, 4050-313 Porto, Portugal

**Keywords:** *Corynebacterium*, genome, 16S rRNA gene, *rpoB*, MALDI-TOF MS, female urinary microbiome

## Abstract

The genus *Corynebacterium* is frequently found in the female urinary microbiome (FUM). In-depth characterization of *Corynebacterium* at the species level has been barely exploited. During ongoing FUM research studies, eight strains (c8Ua_144^T^, c8Ua_172^T^, c8Ua_174^T^, c8Ua_181^T^, c9Ua_112^T^, c19Ua_109^T^, c19Ua_121^T^, and c21Ua_68^T^) isolated from urine samples of healthy women or diagnosed with overactive bladder could not be allocated to any valid *Corynebacterium* species. In this work, we aimed to characterize these strains based on a polyphasic approach. The strains were Gram stain positive, rod to coccoid shaped, nonmotile, catalase positive, and oxidase negative. Phylogenetic analysis based on 16S rRNA and *rpoB* gene sequences indicated that all strains belonged to the genus *Corynebacterium*. The average nucleotide identity and digital DNA–DNA hybridization values among the genomes of the above eight strains and closely related type strains of the *Corynebacterium* genus were <95 (74.1%–93.9%) and <70% (22.2%–56.5%), respectively. Mycolic acids were identified in all strains. MK-8(H2) and/or MK-9(H2) were identified as the major menaquinones. The polar lipids’ pattern mostly consisted of diphosphatidylglycerol, phosphatidylglycerol, and glycophospholipids. The major fatty acid was C_18:1_*ω*9*c*. *Corynebacterium lehmanniae* (c8Ua_144^T^ = DSM 113405^T^ = CCP 74^T^), *Corynebacterium meitnerae* (c8Ua_172^T^ = DSM 113406^T^ = CCP 75^T^), *Corynebacterium evansiae* (c8Ua_174^T^ = DSM 113407^T^ = CCP 76^T^), *Corynebacterium curieae* (c8Ua_181^T^ = DSM 113408^T^ = CCP 77^T^), *Corynebacterium macclintockiae* (c9Ua_112^T^ = DSM 113409^T^ = CCP 78^T^), *Corynebacterium hesseae* (c19Ua_109^T^ = DSM 113410^T^= CCP 79^T^), *Corynebacterium marquesiae* (c19Ua_121^T^ = DSM 113411^T^ = CCP 80^T^), and *Corynebacterium yonathiae* (c21Ua_68^T^ = DSM 113412^T^ = CCP 81^T^) are proposed. This study evidenced that commonly used methodologies on FUM research presented limited resolution for discriminating *Corynebacterium* at the species level. Future research studying the biological mechanisms of the new *Corynebacterium* species here described may shed light on their possible beneficial role for healthy FUM.

## 1. Introduction

The genus *Corynebacterium*, belonging to the family *Corynebacteriaceae* and phylum *Actinobacteria*, was proposed by Lehmann and Neumann in 1896 [1]. Currently, the genus includes 140 species with validly published and correct names under the International Code of Nomenclature of Prokaryotes (https://lpsn.dsmz.de/genus/corynebacterium, accessed on 1 January 2023).

Species of the genus *Corynebacterium* are considered inhabitants of human and animal skin and mucous membranes, and can be found in diverse environmental habitats (e.g., water, soil, and sewage) [2,3]. Nevertheless, some members are increasingly recognized as opportunistic pathogens, being associated with various infections, including urinary tract, soft tissue, central line, and orthopedic infections, as well as bacteremia, pneumonia, septicemia, endocarditis, and meningitis [4,5,6,7].

Recently, *Corynebacterium* has been identified as a frequent and prevalent genus within the healthy and dysbiotic female urinary microbiome, including overactive bladder (OAB) [8,9,10,11]. However, detailed characterization at the species level of *Corynebacterium* has not been addressed, hampering the diversity and potential role of *Corynebacterium* in the healthy and dysbiotic urinary microbiome. During an ongoing project on the female urinary microbiome (FUM) conducted in our research laboratory [9], eight strains representing putative new species of the *Corynebacterium* genus were found. In this study, we aim to characterize and describe these new species that compose the urinary microbiota of healthy women and female patients diagnosed with OAB based on a polyphasic approach.

## 2. Materials and Methods

### 2.1. Participants and Sample Collection

Urine samples were collected from 20 reproductive-aged women and 6 female patients diagnosed with OAB (pre- and postmenopausal women) (2016–2018) [12]. All healthy women followed strict criteria: no pregnancy, no symptoms or diagnosis of current urinary tract infections, and no antibiotic exposure in the previous month. Considering physiological changes ongoing within the female genital tract during the menstrual cycle and the possibility that it could influence the composition of the urinary microbiome, samples were always collected in the same phase of the menstrual cycle (third week).

All women with OAB symptoms, assessed with the Overactive Bladder Symptom Score [13], followed strict criteria: no current UTI (based on urinalysis and standard urine culture), no antibiotic exposure in the past 4 weeks, no pregnancy, and no history of pelvic radiotherapy, bladder tumor, urolithiasis, and neurogenic voiding dysfunction.

Each participant provided a first-morning midstream voided urine sample by a self-performed noninvasive procedure via 40 mL sterile containers.

### 2.2. Culture Conditions and Strain Isolation

Urine samples were processed, up to 2 h after collection, as previously described [10]. Briefly, 100 µL of urine samples were inoculated on Columbia agar with 5% sheep blood ((BAP) Biogerm, Moreira, Portugal) and HiCrome UTI agar (chromogenic agar plates (CAP), HiMedia, India). Colonies were selected after 48 h at 37 °C under aerobic and microaerophilic conditions (BAP and CAP) and anaerobic conditions (BAP). Strains were maintained on tryptic soy broth (TSB; Sigma-Aldrich, St. Louis, MO, USA) supplemented with 30% (*v*/*v*) glycerol at −80 °C for long-term storage. Strains were preliminary identified by a matrix-assisted laser desorption/ionization time-of-flight mass spectrometry (MALDI-TOF) VITEK MS system (bioMérieux, Craponne, France) using in vitro diagnostic database version 3.0. Eight *Corynebacterium* strains (*n* = 7 from urine of healthy women: c8Ua_144^T^, c8Ua_172^T^, c8Ua_174^T^, c8Ua_181^T^, c9Ua_112^T^, c19Ua_109^T^, and c19Ua_121^T^; *n* = 1 from urine of a woman diagnosed with OAB: c21Ua_68^T^) identified as putative new *Corynebacterium* species were further characterized.

### 2.3. Phylogenetic Analysis

Complete nucleotide sequences of the 16S rRNA gene of all the strains analyzed in this study and type strains of closely related *Corynebacterium* species were extracted with MyDbFinder 2.0 (https://cge.food.dtu.dk/services/MyDbFinder/, accessed on 1 April 2022) after genome annotation. Primers and protocols for *rpoB* amplification and sequencing were the same as those described by Khamis et al. (2004) [14]. Further, 16S rRNA and *rpoB gene sequences* were aligned, and similarity scores were generated using MEGA version 7.0 (https://www.megasoftware.net/download_form, accessed on 1 April 2022) [15]. Phylogenetic trees were constructed according to the neighbor-joining method [16], and genetic distances were estimated using Kimura’s two-parameter model [17]. The reliability of internal branches was assessed from bootstrapping based on 1000 resamplings [18].

### 2.4. Comparative Genomic Analysis

Genomic DNA from the eight strains was extracted by the Wizard^®^ Genomic DNA Purification Kit (Promega, Madison, WI, USA). Whole-genome sequencing was performed with the Illumina NovaSeq technology (2 × 150 nt) (Eurofins Scientific, Konstanz, Germany). Illumina reads were trimmed by Trim Galore (http://www.bioinformatics.babraham.ac.uk/projects/trim_galore/, accessed on 1 April 2022), and quality was checked using FastQC v0.11.9 (www.bioinformatics.babraham.ac.uk/projects/fastqc/, accessed on 1 April 2022). De novo assembly was performed by Unicycler v0.4.8 [19], and quality assessed by QUAST v5.0.2 [20]. Annotation of the draft genome was provided by the NCBI Prokaryotic Genome Annotation Pipeline [21]. The average nucleotide identity (ANI) based on BLAST+ was calculated by JspeciesWS [22] and digital DNA–DNA hybridization (dDDH) by the Genome-to-Genome Distance Calculator following the recommended Formula 2 [23].

### 2.5. Phenotypic and Chemotaxonomic Analysis

Growth was evaluated in different culture media: tryptic soy agar (Liofilchem, Roseto degli Abruzzi, Italy), tryptic soy broth (Liofilchem, Italy), brain heart infusion agar (Liofilchem, Italy) and brain heart infusion broth (VWR International, Leuven, Belgium) supplemented or not with 0.1% Tween-80 (Sigma-Aldrich, St. Quentin Fallavier, France) under aerobic conditions, and Columbia agar with 5% sheep blood (bioMérieux, France) under aerobic, microaerophilic, anaerobic conditions and a 5% CO_2_-enriched atmosphere at 37 °C for 48 h. Growths at different temperatures (4, 8, 15, 25, 37, 42, and 50 °C) and NaCl concentrations (5.0, 6.5, 7.0, 7.5, 8.0, and 9.0) were tested in brain heart infusion with 0.1% Tween-80 under aerobic conditions at 37 °C for 48 h. Cell and colony morphology were observed with cells grown on Columbia agar with 5% sheep blood medium at 37 °C for 48 h under aerobic conditions. Gram staining was assessed using a Gram-staining kit (bioMérieux, France). Catalase and oxidase activities were evaluated in the presence of 3% (*v*/*v*) aqueous hydrogen peroxide solution and oxidase strips (Oxoid), respectively. The metabolic profile of the strains was determined by using the API^®^ Coryne and API^®^ Rapid ID 32A bacterial identification systems, according to the instructions of the manufacturer (bioMérieux, France).

Cellular fatty acids were analyzed after conversion into fatty acid methyl esters by saponification, methylation, and extraction using the standard protocol of the Microbial Identification System (MIDI Inc. version 6.1) [24]. To resolve summed features and to confirm the identification from the MIDI system, the samples were additionally analyzed via GC–MS as described previously [25].

Mycolic acids were extracted from approximately 300 mg wet biomass using minor modifications of the method described by Vilchèze and Jacobs (2007) for analysis of mycolic acids by high performance liquid chromatography [26]. Briefly, cells were lysed in 50% KOH–methanol solution (1:1 *v*/*v*) at 95 °C overnight and extracted with chloroform. Dried extracts were reconstituted in chloroform–methanol (9:1) and analyzed on an Agilent 6545 Q-TOF mass spectrometer as described previously [27]. Mycolic acids were identified based on a comparison of exact masses from known mycolic acid structures [28].

Polar lipids were extracted from freeze-dried material based on the method of Bligh and Dyer [29] with slight modifications described by Tindal et al. [30], and separated by two-dimensional thin layer chromatography. Total lipids were visualized by spraying with dodecamolybdophosphoric acid; specific functional groups were visualized with α-naphthol, ninhydrin, or molybdenum blue.

Respiratory quinones were extracted from a freeze-dried cell, as described previously [25]. The separation and identification of quinones were performed by HPLC coupled to a diode array detector and high-resolution mass spectrometer [31].

## 3. Results

### 3.1. MALDI-TOF MS Identification

The strains c8Ua_144^T^, c8Ua_172^T^, c8Ua_181^T^, and c19Ua_109^T^ were initially identified as *Corynebacterium* sp., c8Ua_174^T^ and c9Ua_112^T^ as *Corynebacterium jeikeium*, and c19Ua_121^T^ and c21Ua_68^T^ as *Corynebacterium tuberculostearicum*.

### 3.2. Phylogenetic Analyses Based on 16S rRNA and rpoB Genes

A phylogenetic tree based on 16S rRNA gene sequences showed that all the strains belonged to the genus *Corynebacterium* (Appendix A). The strains c8Ua_181^T^, c21Ua_68^T^, and c19Ua_121^T^ clustered together, and the closest neighbor was *Corynebacterium accolens* JCM 8331^T^ (98.9%–99.2% sequence similarity). The strain c19Ua_109^T^ clustered with *Corynebacterium aurimucosum* IMMIB D-1488 ^T^ (100% sequence similarity), c8Ua_174^T^ and c9Ua_112^T^ with *Corynebacterium jeikeium* JCM 9384^T^ (100% and 99.8% sequence similarity, respectively), c8Ua_144^T^ with *Corynebacterium fournieri* Marseille-P2948 ^T^ (99.4% sequence similarity), and c8Ua_172^T^ with *Corynebacterium tuscaniense* ISS-5309 ^T^ (99.8% sequence similarity) (Appendix A).

The topology of a partial *rpoB*-based phylogenetic tree revealed few differences compared with the 16S rRNA gene (Appendix A), yet the nucleotide sequence similarities were quite different. The type strain of *C. tuberculostearicum* was the closest neighbor of c8Ua_181^T^, c19Ua_121^T^, and c21Ua_68^T^ (94.8%, 95.2% and 95.8% sequence similarity, respectively), while the type strain of *Corynebacterium afermentans* was the closest neighbor of c8Ua_144^T^ (97.0% sequence similarity). The strains c8Ua_174^T^ and c9Ua_112^T^ clustered with *C. jeikeium* (95.5% and 95.8% sequence similarity, respectively), and the closest neighbor of the strains c8Ua_172^T^ and c19Ua_109^T^ was *C. tuscaniense* (90.8% sequence similarity) and *C. aurimucosum* (95.5% sequence similarity), respectively.

### 3.3. Genomic Analysis

The genomic features of the strains analyzed in this study and the ANI and dDDH values between our strains and the closest related type strains of the *Corynebacterium* species are presented in Table 1.

The ANI and dDDH values between the eight strains were ≤66.3% and 35.9%, respectively. The ANI values between the strains c8Ua_181^T^, c19Ua_121^T^, and c21Ua_68^T^ and the closest related *C. tuberculostearicum* were 87.7%, 94.0%, and 88.6%, respectively (Table 1). The ANI values between c8Ua_174^T^ or c9Ua_112^T^ and *C. jeikeium* were 91.4% and 85.7%, respectively (Table 1). The ANI values between the strains c8Ua_144^T^, c8Ua_172^T^, or c19Ua_109^T^ and the closest species *C. afermentans*, *C. tuscaniense*, and *C. aurimucosum* were 90.3, 84.1, and 88.4, respectively (Table 1). Likewise, the dDDH values between the eight strains and between all the strains analyzed and the closest *Corynebacterium* species ranged from 28.6% to 55.6% (Table 1).

Remarkably, 46 available genomes in public databases from strains deposited as *Corynebacterium* sp., *C. jeikeium*, *C. aurimucosum*, *C. tuberculostearicum*, and not validly published species (e.g., *Corynebacterium haemomassilienses*) should be reclassified based on whole-genome relatedness, since ANI values between these strains and those analyzed in this study were all above 95% (Appendix A). In fact, the ANI values between 27 publicly available genomes from strains deposited as *Corynebacterium* sp. or *C. aurimucosum*, isolated from *Homo sapiens* or environmental samples, and *Corynebacterium hessae* c19Ua_109^T^ ranged from 96.0% to 96.8%. The ANI values between 11 strains deposited as *Corynebacterium* sp. or *C. jeikeium*, isolated from *Homo sapiens* samples, and *Corynebacterium macclintockiae* c9Ua_112^T^ ranged from 96.8% to 97.6%. The ANI values between four strains deposited as *C. aurimucosum* or *C. tuberculostearicum*, isolated from *Homo sapiens*, and *Corynebacterium marquesiae* c19Ua_121^T^ ranged from 95.9% to 96.6%. Two strains (*Corynebacterium* sp. and *C. jeikeium*), isolated from *Homo sapiens*, showed ANI values of 97.0% and 97.1% with *Corynebacterium evansiae* c8Ua_174^T^, respectively. Finally, two other strains isolated from *Homo sapiens*, and identified as *Corynebacterium haemomassilienses* or *Corynebacterium* sp., showed ANI values of 96.1% and 99.9% with *Corynebacterium lehmanniae* c8Ua_144^T^ and *Corynebacterium yonathiae* c21Ua_68^T^, respectively.

### 3.4. Phenotypic Characterization

The strains grow in all the tested medium under aerobic conditions. Supplementation of the tested culture medium with Tween-80 allowed better growth of all but one strain (c19Ua_109^T^). Optimum growth was observed for all strains growing in Columbia agar with 5% sheep blood under an aerobic and 5% CO_2_-enriched atmosphere, at 37 °C for 48 h. Good growth was observed in microaerophilic conditions, and variable growth was reported in anaerobic conditions. Bacterial cells appeared as Gram stain positive, rod to coccoid shaped, no longer than 1.75 µm in length. All strains were catalase positive, oxidase negative, and nonmotile. The differential characteristics from other species are listed in Table 2.

### 3.5. Chemotaxonomic Characterizations

Short-chain mycolic acids with 32–38 carbons were detected in all strains. The strains c8Ua_144^T^, c8Ua_174^T^, c8Ua_181^T^, c9Ua_112^T^, and c19Ua_109^T^ contained MK-9(H2) as the major menaquinone (65.1%, 76.6%, 51.6%, 80.3%, 51.1%, and 60.0%, respectively), while the strains c8Ua_172^T^, c19Ua_121^T^, and c21Ua_68^T^ contained MK-8(H2) (51.7%, 60.4%, and 49.4%, respectively).

The polar lipid analysis detected the presence of diphosphatidylglycerol, phosphatidylglycerol, as well as uncharacterized phospholipids, and glycolipids in all strains. Additionally, glycophospholipids were detected in all but one strain (c8Ua_174^T^), uncharacterized aminolipids were detected in two strains (c19Ua_109^T^ and c19Ua_121^T^), and uncharacterized lipids were detected in all but two strains (c8Ua_181^T^ and c21Ua_68^T^) (Appendix A).

The cellular fatty acid profile of the analyzed strains was composed predominantly of straight-chain, saturated, and monounsaturated fatty acids, with the predominance of cis-9-octadecenoic acid (oleic acid, C_18:1_
*ω*9*c*), followed by moderate amounts of hexadecanoic acid (palmitic acid, C_16:0_) in all strains (Table 3). Moreover, tuberculostearic acid (10-methyl-C_18:0_) was detected in small amounts in five strains.

### 3.6. Description of Corynebacterium sp. nov.

The proposed names of *Corynebacterium* novel species described in this study are in honor of women who changed the world or female researchers whose role in science was neglected and only later received due credit for their discoveries.

*Corynebacterium lehmanniae* sp. nov.

*Corynebacterium lehmanniae* (leh.man’ni.ae. N.L. gen. fem. n. *lehmanniae* of Lehman; in honor of Inge Lehmann, a Danish seismologist, who discovered that Earth has a solid inner core).

Cells are irregular rod and Gram stain positive with a mean length of 1.5 µm. Colonies are smooth and slightly grayish, small, with a mean diameter of 1 mm on Columbia agar with 5% sheep blood. Optimum growth was observed at 37 °C for 48 h under aerobic conditions or a 5% CO_2_-enriched atmosphere on Columbia agar with 5% sheep blood. It grows well in brain heart infusion and tryptic soy medium supplemented with 0.1% Tween-80. Growth is observed between 20 and 37 °C and NaCl at 8.0% (*w*/*v*) maximum. Cells are catalase positive and oxidase negative. In API^®^ Coryne and API^®^ Rapid ID 32A, acid is not produced from D-glucose, D-ribose, D-xylose, D-mannitol, D-maltose, D-lactose, D-saccharose, D-mannose, D-raffinose, and glycogen. Positive enzymatic reaction is observed for proline arylamidase. Cells are negative for nitrate, gelatin, esculin, indole, urease, arginine dihydrolase, α-galactosidase, β-galactosidase, β-galactosidase-6-phosphate, α-glucosidase, β-glucosidase, α-arabinosidase, β-glucuronidase, N-acetyl-β-glucosaminidase, glutamic acid decarboxylase, α-fucosidase, alkaline phosphatase, arginine arylamidase, leucyl glycine arylamidase, phenylalanine arylamidase, leucine arylamidase, pyroglutamic acid arylamidase, tyrosine arylamidase, alanine arylamidase, glycine arylamidase, histidine arylamidase, glutamyl glutamic acid, arylamidase, serine arylamidase, pyrazinamidase, and pyrrolidonyl arylamidase. Corynemycolic acids are present. Polar lipid analysis revealed the presence of diphosphatidylglycerol, phosphatidylglycerol, and uncharacterized glycophospholipids, glycolipids, phospholipid, and lipids. The quinone system is characterized by the major menaquinone MK-9(H2), and the major fatty acid is C_18:1_
*ω*9*c*.

The type strain, c8Ua_144^T^ (=DSM 113405^T^ = CCP 74^T^), was isolated from the urine of a healthy woman in Portugal in 2017. Another strain was isolated from blood (Appendix A). The DNA G + C content of the type strain is 65.4 mol %. The annotated genomic sequence of the strain c8Ua_144^T^ was deposited in DDBJ/ENA/GenBank and is available under accession number JAKMUR000000000.

*Corynebacterium meitnerae* sp. nov.

*Corynebacterium meitnerae* (mei’tner.ae. N.L. gen. fem. n. *meitnerae* of Meitner; in honor of Lise Meitner, an Austrian physicist, who studied radioactivity and nuclear physics and, together with Otto Hahn and Fritz Straßmann, discovered the nuclear fission).

Cells are coccoid or irregular rod and Gram stain positive with a mean length of 1.02 µm. Colonies are whitish, with entire edges, no-hemolytic, and very small (≤1 mm) on Columbia agar with 5% sheep blood. Optimum growth is observed at 37 °C for 48 h under aerobic conditions or a 5% CO_2_-enriched atmosphere on Columbia agar with 5% sheep blood. It grows well in brain heart infusion and tryptic soy medium supplemented with 0.1% Tween-80. Growth is observed between 15 and 42 °C and NaCl at 7.0% (*w*/*v*) maximum. Cells are catalase positive and oxidase negative. In API^®^ Coryne and API^®^ Rapid ID 32A, acid is not produced from D-glucose, D-ribose, D-xylose, D-mannitol, D-maltose, D-lactose, D-saccharose, D-mannose, D-raffinose, and glycogen. Positive enzymatic reaction is observed for alkaline phosphatase. Cells are negative for nitrate, gelatin, esculin, indole, urease, arginine dihydrolase, α-galactosidase, β-galactosidase, β-galactosidase-6-phosphate, α-glucosidase, β-glucosidase, α-arabinosidase, β-glucuronidase, N-acetyl-β-glucosaminidase, glutamic acid decarboxylase, α-fucosidase, leucyl glycine arylamidase, leucyl arylamidase, phenylalanine arylamidase, leucine arylamidase, pyroglutamic acid arylamidase, proline arylamidase, tyrosine arylamidase, alanine arylamidase, glycine arylamidase, histidine arylamidase, glutamyl glutamic acid, arylamidase, serine arylamidase, pyrazinamidase, and pyrrolidonyl arylamidase. Corynemycolic acids are present. Polar lipid analysis revealed the presence of diphosphatidylglycerol, phosphatidylglycerol, and uncharacterized glycophospholipids, glycolipids, phospholipids, and lipids. The quinone system is characterized by the major menaquinone MK-8(H2), and the major fatty acid is C_18:1_
*ω*9*c*.

The type strain, c8Ua_172^T^ (=DSM 113406^T^ = CCP 75^T^), was isolated from the urine of a healthy woman in Portugal in 2017. The DNA G + C content of the type strain is 61.0 mol %. The annotated genomic sequence of the strain c8Ua_172^T^ was deposited in DDBJ/ENA/GenBank and is available under accession number JAKMUS000000000.

*Corynebacterium evansiae* sp. nov.

*Corynebacterium evansiae* (e.van’si.ae. N.L. gen. fem. n. evansiae of Evans; in honor of Alice Evans, an American microbiologist, who demonstrated that *Brucella abortus* caused the disease brucellosis (undulant fever) in both cattle and humans, suggesting that raw milk could cause disease in humans).

Cells are pleomorphic rod and Gram stain positive with a mean length of 0.9 µm. Colonies are greyish to white, with entire edges, small with a mean diameter of 1 mm, and nonhemolytic on Columbia agar with 5% sheep blood. Optimum growth is observed at 37 °C for 48 h under aerobic conditions or a 5% CO_2_-enriched atmosphere on Columbia agar with 5% sheep blood. It grows well in brain heart infusion and tryptic soy medium supplemented with 0.1% Tween-80. Growth occurs between 15 and 42 °C and NaCl at 9.0% (*w*/*v*) maximum. Cells are catalase positive and oxidase negative. In API^®^ Coryne and API^®^ Rapid ID 32A, acid is produced from D-glucose (weak reaction). Acid is not produced from D-ribose, D-xylose, D-mannitol, D-maltose, D-lactose, D-saccharose, D-mannose, D-raffinose, and glycogen. Positive enzymatic reaction is observed for alkaline phosphatase, arginine arylamidase, and phenylalanine arylamidase and weakly positive for glycine arylamidase and pyrazinamidase. Cells are negative for nitrate, gelatin, esculin, indole, urease, arginine dihydrolase, α-galactosidase, β-galactosidase, β-galactosidase-6-phosphate, α-glucosidase, β-glucosidase, α-arabinosidase, β-glucuronidase, N-acetyl-β-glucosaminidase, glutamic acid decarboxylase, α-fucosidase, leucyl glycine arylamidase, leucine arylamidase, pyroglutamic acid arylamidase, proline arylamidase, tyrosine arylamidase, alanine arylamidase, histidine arylamidase, glutamyl glutamic acid, arylamidase, serine arylamidase, and pyrrolidonyl arylamidase. Corynemycolic acids are present. Polar lipid analysis revealed the presence of diphosphatidylglycerol phosphatidylglycerol and uncharacterized glycolipids, phospholipids and lipids. The quinone system is characterized by the major menaquinone MK-9(H2), and the major fatty acid is C_18:1_
*ω*9*c*.

The type strain, c8Ua_174^T^ (=DSM 113407^T^ = CCP 76^T^), was isolated from the urine of a healthy woman in Portugal in 2017. Other strains were isolated from vertebral disk space and urine (Appendix A). The DNA G + C content of the type strain is 62.9 mol %. The annotated genomic sequence of the strain c8Ua_174^T^ was deposited in DDBJ/ENA/GenBank and is available under accession number JAKMUT000000000.

*Corynebacterium curieae* sp. nov.

*Corynebacterium curieae* (cu.rie’ae. N.L. gen. fem. n. *curieae* of Curie; in honor of Marie Skłodowska-Curie, a Polish physicist and chemist, who conducted pioneering research on radioactivity).

Cells are irregular rods to coccoid and Gram stain positive with a mean length of 1.10 µm. Colonies are almost colorless, very small (<1 mm in diameter), and nonhemolytic on Columbia agar with 5% sheep blood. Optimum growth is observed at 37 °C for 48 h under aerobic conditions or a 5% CO_2_-enriched atmosphere on Columbia agar with 5% sheep blood. It grows well in brain heart infusion and tryptic soy medium supplemented with 0.1% Tween-80. Growth is observed between 25 and 42 °C and NaCl at 9.0% (*w*/*v*) maximum. Cells are catalase positive and oxidase negative. In API^®^ Coryne and API^®^ Rapid ID 32A, acid is produced from D-mannose. There is weak reaction for D-glucose and D-ribose. Acid is not produced from D-xylose, D-mannitol, D-maltose, D-lactose, D-saccharose, D-raffinose, and glycogen. Positive enzymatic reaction is observed for alkaline phosphatase and proline arylamidase. Cells are negative for nitrate, gelatin, esculin, indole, urease, arginine dihydrolase, α-galactosidase, β-galactosidase, β-galactosidase-6-phosphate, α-glucosidase, β-glucosidase, α-arabinosidase, β-glucuronidase, N-acetyl-β-glucosaminidase, glutamic acid decarboxylase, α-fucosidase, leucyl glycine arylamidase, leucine arylamidase, pyroglutamic acid arylamidase, tyrosine arylamidase, alanine arylamidase, histidine arylamidase, glutamyl glutamic acid, arylamidase, serine arylamidase, pyrrolidonyl arylamidase, phenylalanine arylamidase, glycine arylamidase, and pyrazinamidase. Corynemycolic acids are present. Polar lipid analysis revealed the presence of diphosphatidylglycerol, phosphatidylglycerol, and uncharacterized phospholipids, glycophospholipid and glycolipids. The quinone system is characterized by the major menaquinone MK-9(H2), and the major fatty acid is C_18:1_
*ω*9*c*.

The type strain, c8Ua_181^T^ (=DSM 113408^T^ = CCP 77^T^), was isolated from the urine of a healthy woman in Portugal in 2017. The DNA G + C content of the type strain is 58.5 mol %. The annotated genomic sequence of the strain c8Ua_181^T^ was deposited in DDBJ/ENA/GenBank and is available under accession number JAKMUU000000000.

*Corynebacterium macclintockiae* sp. nov.

*Corynebacterium macclintockiae* (mcclin.tock’i.ae. N.L. gen. fem. n. *mcclintockiae* of McClintock; in honor of Barbara McClintock, an American cytogeneticist, who discovered transposons).

Cells are irregular rods to coccoid and Gram stain positive with a mean length of 1.0 µm. Colonies are greyish white, with entire edges, small with a mean diameter of 1.0 mm, and nonhemolytic on Columbia agar with 5% sheep blood. Optimum growth is observed at 37 °C for 48 h under aerobic conditions or a 5% CO_2_-enriched atmosphere on Columbia agar with 5% sheep blood. It grows well in brain heart infusion and tryptic soy medium supplemented with 0.1% Tween-80. Growth is observed between 25 and 42 °C and NaCl at 7.5% (*w*/*v*) maximum. Cells are catalase positive and oxidase negative. In API^®^ Coryne and API^®^ Rapid ID 32A, acid is not produced from D-glycose, D-ribose, D-xylose, D-mannitol, D-maltose, D-lactose, D-saccharose, D-mannose, D-raffinose, and glycogen. Positive enzymatic reaction is observed for alkaline phosphatase. Cells are negative for nitrate, gelatin, esculin, indole, urease, arginine dihydrolase, α-galactosidase, β-galactosidase, β-galactosidase-6-phosphate, α-glucosidase, β-glucosidase, α-arabinosidase, β-glucuronidase, N-acetyl-β-glucosaminidase, glutamic acid decarboxylase, α-fucosidase, leucyl glycine arylamidase, leucine arylamidase, pyroglutamic acid arylamidase, proline arylamidase, tyrosine arylamidase, alanine arylamidase, histidine arylamidase, glutamyl glutamic acid arylamidase, and serine arylamidase and pyrrolidonyl arylamidase, arginine arylamidase, phenylalanine arylamidase, glycine arylamidase, and pyrazinamidase. Corynemycolic acids are present. Polar lipid analysis revealed the presence of diphosphatidylglycerol, phosphatidylglycerol, and uncharacterized glycolipids, glycophospholipid, phospholipid, and lipids. The quinone system is characterized by the major menaquinone MK-9(H2), and the major fatty acid is C_18:1_
*ω*9*c*.

The type strain, c9Ua_112^T^ (=DSM 113409^T^ = CCP 78^T^), was isolated from the urine of a healthy woman in Portugal in 2017. Other strains were isolated from pleural human spine, knee bursa, humeral membrane, tissue, diabetic foot ulcer, and decubitus ulcer (Appendix A). The DNA G + C content of the type strain is 62.0 mol %. The annotated genomic sequence of the strain c9Ua_112^T^ was deposited in DDBJ/ENA/GenBank and is available under accession number JAKMUV000000000.

*Corynebacterium hesseae* sp. nov.

*Corynebacterium hesseae* (hes’se.ae. N.L. gen. fem. n. *hesseae* of Hesse; in honor of Fanny Hesse, an American microbiologist, who alongside her husband, Walther Hesse, pioneered the use of agar as a common gelling agent for the production of media capable of culturing microorganisms).

Cells are irregular rods and Gram stain positive with a mean length of 1.4 µm. Colonies are whitish to beige, with entire edges with 1–2 mm in diameter, and hemolytic on Columbia agar with 5% sheep blood. Optimum growth is observed at 37 °C for 48 h under aerobic conditions or a 5% CO_2_-enriched atmosphere on Columbia agar with 5% sheep blood. Growth is observed between 15 and 42 °C and NaCl at 8.0% (*w*/*v*) maximum. Cells are catalase positive and oxidase negative. In API^®^ Coryne and API^®^ Rapid ID 32A, acid is produced from D-glycose, D-maltose, and D-mannose. Acid is not produced from D-ribose, D-xylose, D-mannitol, D-lactose, D-saccharose, D-mannose, D-raffinose, and glycogen. Positive enzymatic reaction is observed for proline arylamidase, arginine arylamidase, alanine arylamidase, glycine arylamidase, and pyrazinamidase. Cells are negative for nitrate, gelatin, esculin, indole, urease, arginine dihydrolase, α-galactosidase, β-galactosidase, β-galactosidase-6-phosphate, α-glucosidase, β-glucosidase, α-arabinosidase, β-glucuronidase, N-acetyl-β-glucosaminidase, glutamic acid decarboxylase, α-fucosidase, alkaline phosphatase, leucyl glycine arylamidase, leucine arylamidase, pyroglutamic acid arylamidase, proline arylamidase, tyrosine arylamidase, histidine arylamidase, glutamyl glutamic acid arylamidase, and serine arylamidase and pyrrolidonyl arylamidase and phenylalanine arylamidase. Corynemycolic acids are present. Polar lipid analysis revealed the presence of diphosphatidylglycerol, phosphatidylglycerol, and uncharacterized glycolipids, phospholipids, glycophospholipid, aminolipids, and lipids. The quinone system is characterized by the major menaquinone MK-9(H2), and the major fatty acid is C_18:1_
*ω*9*c*.

The type strain, c19Ua_109^T^ (=DSM 113410^T^ = CCP 79^T^), was isolated from the urine of a healthy woman in Portugal in 2017. Other strains were isolated from a human (vagina, rectum, urine, sebaceous cyst, skin, nasal mucosa) and the environment. The DNA G + C content of the type strain is 61.0 mol %. The annotated genomic sequence of the strain c19Ua_109^T^ was deposited in DDBJ/ENA/GenBank and is available under accession number JAKMUW000000000.

*Corynebacterium marquesiae* sp. nov.

*Corynebacterium marquesiae* (mar.que’si.ae. N.L. gen. fem. n. *marquesiae* of Marques; in honor of Branca Edmée Marques, a Portuguese scientist, who worked under the famous scientist Madame Marie Curie, and whose research in nuclear physics for peaceful means was finally recognized by the Portuguese State at the age of 67).

Cells are irregular rods to coccoid non-spore-forming with a mean length of 1.26 µm. Colonies are slightly white, very small (≤1 mm in diameter), and nonhemolytic on Columbia agar with 5% sheep blood. Optimum growth is observed at 37 °C for 48 h under aerobic conditions or a 5% CO_2_-enriched atmosphere on Columbia agar with 5% sheep blood. It grows well in brain heart infusion and tryptic soy medium supplemented with 0.1% Tween-80. Growth is observed between 15 and 42 °C and NaCl at 9.0% (*w*/*v*) maximum. Cells are catalase positive and oxidase negative. In API^®^ Coryne and API^®^ Rapid ID 32A, acid is not produced from D-glucose, D-ribose, D-xylose, D-mannitol, D-maltose, D-lactose, D-saccharose, D-raffinose, D-mannose, and glycogen. Cells are negative for nitrate, gelatin, esculin, and indole tests. Positive enzymatic reaction is observed for pyroglutamic acid arylamidase and proline arylamidase. Cells are negative for urease, arginine dihydrolase, α-galactosidase, β-galactosidase, β-galactosidase-6-phosphate, α-glucosidase, β-glucosidase, α-arabinosidase, β-glucuronidase, N-acetyl-β-glucosaminidase, glutamic acid decarboxylase, α-fucosidase, leucyl glycine arylamidase, leucine arylamidase, tyrosine arylamidase, alanine arylamidase, histidine arylamidase, glutamyl glutamic acid arylamidase, and serine arylamidase and pyrrolidonyl arylamidase, phenylalanine arylamidase, glycine arylamidase, alkaline phosphatase, and pyrazinamidase. Corynemycolic acids are present. Polar lipid analysis revealed the presence of diphosphatidylglycerol, phosphatidylglycerol, and uncharacterized glycolipids, phospholipid, glycophospholipid, aminolipids, and lipids. The quinone system is characterized by the major menaquinone MK-8(H2), and the major fatty acid is C_18:1_
*ω*9*c*.

The type strain, c19Ua_121^T^ (=DSM 113411^T^ = CCP 80^T^), was isolated from the urine of a healthy woman in Portugal in 2017. Other strains were isolated from the skin (Appendix A). The DNA G + C content of the type strain is 59.5 mol %. The annotated genomic sequence of the strain c19Ua_121^T^ was deposited in DDBJ/ENA/GenBank and is available under accession number JAKMUY000000000.

*Corynebacterium yonathiae* sp. nov.

*Corynebacterium yonathiae* (yo.na’thi.ae. N.L. gen. fem. n. *yonathiae* of Yonath; in honor of Ada Yonath, an Israeli chemistry, for her studies on the structure and function of the ribosome.

Cells are pleomorphic rods to coccoid and Gram stain positive with a mean length of 1.33 µm. Colonies are circular, whitish, small with a mean diameter of 1 mm, and nonhemolytic on Columbia agar with 5% sheep blood. Optimum growth is observed at 37 °C for 48 h under aerobic conditions or a 5% CO_2_-enriched atmosphere on Columbia agar with 5% sheep blood. It grows well in brain heart infusion and tryptic soy medium supplemented with 0.1% Tween-80. Growth is observed between 8 and 42 °C and NaCl at 8.0% (*w*/*v*) maximum. Cells are catalase positive and oxidase negative. In API^®^ Coryne and API^®^ Rapid ID 32A, acid is not produced from D-glucose, D-ribose, D-xylose, D-mannitol, D-maltose, D-lactose, D-saccharose, D-raffinose, D-mannose, and glycogen. Positive enzymatic reaction is observed for proline arylamidase and alkaline phosphatase. Cells are negative for nitrate, gelatin, esculin, indole, urease, arginine dihydrolase, α-galactosidase, β-galactosidase, β-galactosidase-6-phosphate, α-glucosidase, β-glucosidase, α-arabinosidase, β-glucuronidase, N-acetyl-β-glucosaminidase, glutamic acid decarboxylase, α-fucosidase, leucyl glycine arylamidase, leucine arylamidase, pyroglutamic acid arylamidase, tyrosine arylamidase, alanine arylamidase, histidine arylamidase, glutamyl glutamic acid arylamidase, and serine arylamidase and pyrrolidonyl arylamidase, phenylalanine arylamidase, glycine arylamidase, and pyrazinamidase. Corynemycolic acids are present. Polar lipid analysis revealed the presence of diphosphatidylglycerol, phosphatidylglycerol, and uncharacterized glycolipids, phospholipid, and glycophospholipid. The quinone system is characterized by the major menaquinone MK-8(H2), and the major fatty acid is C_18:1_
*ω*9*c*.

The type strain, c21Ua_68^T^ (=DSM 113412^T^ = CCP 81^T^), was isolated from the urine of a woman with overactive bladder symptoms in Portugal in 2017. Another strain was isolated from nasopharynx (Appendix A). The DNA G + C content of the type strain is 58.7 mol %. The annotated genomic sequence of the strain c21Ua_68^T^ was deposited in DDBJ/ENA/GenBank and is available under accession number JAKMUZ000000000.

## 4. Discussion

The accurate discrimination of the *Corynebacterium* species has been improved over the years with the emergence of bacterial identification tools. However, the identification of some closest species is still a challenge. MALDI-TOF MS is a powerful tool to rapidly and accurately identify bacteria to both genus and species levels [37,38], yet it failed to differentiate the strains analyzed in this study.

The 16S rRNA gene is the most widely used marker to determine phylogenetic relationships of bacteria. However, the 16S rRNA gene provides limited resolution to discriminate among closely related species of the genus *Corynebacterium* due to a low polymorphism degree [39,40]. Our study also corroborates this since all but one *Corynebacterium* strains analyzed clustered with other *Corynebacterium* species. Conversely, some studies have shown that the partial sequence of *rpoB* is more accurate than the 16S rRNA gene to discriminate closely related *Corynebacterium* species [40,41]. Khamis and colleagues (2005) observed that the highest degree of similarity of *rpoB* partial sequence between two species was 95.0% [40]. However, the in silico analysis performed in this study revealed that the highest degree of similarity of *rpoB* partial sequence between the two most closely related *Corynebacterium* species (i.e., *Corynebacterium diphtheriae* and *Corynebacterium belfantii*) was 98.1%, and most of the strains here analyzed showed high values of similarity degree with the most related species (above the 95.0% cut-off set out by Khamis) [40]. Therefore, neither the 16S rRNA gene nor *rpoB* provides an accurate identification of closely related *Corynebacterium* species.

Whole-genome analysis is the most powerful tool for the taxonomic classification of the genus *Corynebacterium* [38,42]. In the current work, it allowed for the accurate discrimination of all *Corynebacterium* strains analyzed. The ANIb values between our strains and between our strains and closely related *Corynebacterium* species were clearly below the proposed species cut-off of 95%–96% [43]. Likewise, the dDDH values between the strains analyzed and the closest *Corynebacterium* species were below the proposed species cut-off level of 70% [23]. These data confirmed that the eight strains analyzed represent distinct species within the *Corynebacterium* genus. Moreover, the ANI values obtained between our strains and publicly available genomes support the reclassification of 46 strains as *C. hesseae* (27 strains), *C. macclintockiae* (11 strains), *C. marquesiae* (4 strains)*, C. evansiae* (2 strains), *C. lehmanniae*, and *C. yonathiae* (1 strain each).

The morphologic and phenotypic characteristics (e.g., Gram positive staining, pleomorphic, rod to coccoid shaped, nonmotile, non-spore-forming, positive catalase, and oxidase negative) of our strains are in accordance with those previously reported for the *Corynebacterium* genus [38,39,44]. Likewise, the short-chain mycolic acids with 32–38 carbons, the major menaquinone (types MK-8(H2) and MK-9(H2), and the complex polar lipid and cellular fatty acid profiles (predominance of C_18:1_*ω*9*c*) exhibited are also in line with those reported for other species of the genus [39,42,45].

Our study supports that the strains analyzed in this study represent eight novel species within the *Corynebacterium* genus, for which the names *Corynebacterium lehmanniae*, *Corynebacterium meitnerae*, *Corynebacterium evansiae*, *Corynebacterium curieae*, *Corynebacterium macclintockiae*, *Corynebacterium hesseae*, *Corynebacterium marquesiae,* and *Corynebacterium yonathiae* are proposed. However, the biological mechanisms that *Corynebacterium* members composing the FUM may play in maintaining a healthy microbiome remain to be deciphered. Nevertheless, recent studies revealed that some *Corynebacterium* species may play an important role in microbiome-mediated protection against pathogens [46,47,48]. For instance, *Corynebacterium accolens* has been associated with the inhibition of nasal colonization by *Streptococcus pneumoniae* and *Corynebacterium pseudodiphtheriticum* with the eradication of *Staphylococcus aureus* from the human nose, including methicillin-resistant *S. aureus)* [49,50].

## 5. Conclusions

Our study evidenced that 16S RNA gene sequencing, previously proposed cut-off of *rpoB*, as well as MALDI-TOF spectroscopy, presented limited resolution for discriminating closely related and new *Corynebacterium* species, leading to an underestimated diversity of this genus in the female urinary microbiome.

This work expanded the knowledge on the diversity of *Corynebacterium* species inhabiting the FUM. Future research studying the biological mechanisms of the new *Corynebacterium* species here described, in the context of the genitourinary tract, may shed light on their possible beneficial role.

## Figures and Tables

**Table 1 microorganisms-11-00388-t001:** Genomic features of the *Corynebacterium* strains analyzed in this study.

Strain	Species	Number of Contigs	Genome Size (Mbp)	G + C (%)	tRNA	Proteins	CDS	Genomic Accession Number	Closest Related *Corynebacterium* Species	
									ANI (%)	dDDH (%)
c8Ua_144	*Corynebacterium lehmanniae*	81	2.5	65.4	47	2349	2373	JAKMUR000000000	*C. afermentans* DSM 44280^T^	
									90.3	42.6
c8Ua_172	*Corynebacterium meitnerae*	50	2.3	61.0	44	2078	2101	JAKMUS000000000	*C. tuscaniense* CCUG 51321^T^	
									84.7	28.6
c8Ua_174	*Corynebacterium evansiae*	28	2.2	62.9	50	1922	1999	JAKMUT000000000	*C. jeikeium* ATCC 43734^T^	
									91.4	44.2
c8Ua_181	*Corynebacterium curieae*	30	2.5	58.5	51	2282	2316	JAKMUU000000000	*C. tuberculostearicum* DSM 44922^T^	
									87.8	34.4
c9Ua_112	*Corynebacterium macclintockiae*	35	2.2	62.0	56	1931	1974	JAKMUV000000000	*C. jeikeium* ATCC 43734^T^	
									85.6	29.7
c19Ua_109	*Corynebacterium hessae*	32	2.6	61.0	51	2431	2465	JAKMUW000000000	*C. aurimucosum* DSM 44532^T^	
									88.4	34.7
c19Ua_121	*Corynebacterium marquesiae*	34	2.5	59.5	50	2274	2309	JAKMUY000000000	*C. tuberculostearicum* DSM 44922^T^	
									94.0	55.6
c21Ua_68	*Corynebacterium yonathiae*	54	2.4	58.7	50	2232	2270	JAKMUZ000000000	*C. tuberculostearicum* DSM 44922^T^	
									88.7	36.4

**Table 2 microorganisms-11-00388-t002:** Differential characteristics of the strains analyzed in this study and related type strains of *Corynebacterium* species. Strains: 1, c8Ua_144; 2, c8Ua_172; 3, c8Ua_174; 4, c8Ua_181; 5, c9Ua_112; 6, c19Ua_109; 7, c19Ua_121; 8, c21Ua_68; 9, *C. afermentans* CIP 103499 [32]; 10, *C. tuscaniense* CCUG 51321 [33]; 11, *C. jeikeium* ATCC 43734 [34]; 12, *C. tuberculostearicum* DSM 44922 [35]; 13, *C. aurimucosum* DSM 44532 [36].

Characteristic	1	2	3	4	5	6	7	8	9	10	11	12	13
Conditions for growth:													
Temperature (°C)	20–37	15–42	15–42	25–42	25–42	15–42	15–42	15–42	nd	nd	nd	nd	nd
(NaCl) (%)	5–8	5–7	5–9	5–9	5–7.5	5–8	5–9	5–8	5–6.5	nd	nd	nd	nd
Nitrate reduction	−	−	−	−	−	−	−	−	−	−	−	v	−
Esculin hydrolysis	−	−	−	−	−	−	−	−	−	−	−	−	−
Gelatin hydrolysis	−	−	−	−	−	−	−	−	−	−	−	v	−
Fermentation of:													
D-glucose	−	−	w	w	−	+	−	−	−	+	+	+	+
D-mannose	−	−	−	w	−	+	−	−	−	nd	−	−	−
D-ribose	−	−	−	w	−	−	−	−	−	−	nd	+	−
D-maltose	−	−	−	−	−	+	−	−	−	+	v	v	+
D-saccharose	−	−	−	−	−	−	−	−	−	−	−	v	+
Enzyme activity:													
Urease	−	−	−	−	−	−	−	−	−	−	−	−	−
Pyrazinamidase	−	−	w	−	−	+	−	−	+	+	+	+	+
α-Glucosidase	−	−	−	−	−	−	−	−	−	−	−	−	−
ß-Glucosidase	−	−	−	−	−	−	−	−	−	nd	nd	−	−
ß-Glucuronidase	−	−	−	−	−	−	−	−	nd	−	−	nd	−
Alkaline phosphatase	−	+	+	+	+	−	−	−	+	+	+	v	+
Leucine arylamidase	−	−	−	−	−	−	−	−	nd	nd	nd	v	+

+, positive; w, weakly positive; −, negative; v, variable reaction; nd, not determined.

**Table 3 microorganisms-11-00388-t003:** Cellular fatty acid composition (percentages) of the 10 strains analyzed in this study and closely related type strains. Strains: 1, c8Ua_144; 2, c8Ua_172; 3, c8Ua_174; 4, c8Ua_181; 5, c9Ua_112; 6, c19Ua_109; 7, c19Ua_121; 8, c21Ua_68; 9, *C. afermentans* CIP 103499 [32]; 10, *C. tuscaniense* CCUG 51321 [33]; 11, *C. jeikeium* ATCC 43734 [34]; 12, *C. tuberculostearicum* DSM 44922 [35]; 13, *C. aurimucosum* DSM 44532 [36].

Fatty Acid (%) ^1^	1	2	3	4	5	6	7	8	9	10	11	12	13
C_14:0_	-	-	-	-	-	-	-	-	-	nd	1.0	-	1.2
C_15:0_	-	-	-	-	-	-	-	-	-	nd	-	3.0	-
C_16:0_	11.4	26.7	18.9	11.1	17.9	15.3	16.9	13.7	+ (ni)	nd	27.0	28.0	55.5
C_17:0_	-	-	-	-	-	-	-	-	-	nd	2.0	2.0	-
C_18:0_	1.1	16.1	1.6	1.5	tr	2.0	1.6	1.4	-	nd	16.0	14.0	3.2
C_18:1_ ω9c	74.1	37.9	58.8	74.8	61.4	68.8	66.1	74.4	+ (ni)	nd	28.0	26.0	32.6
10-Methyl-C_18:0_	6.5	-	-	5.6	-	4.7	5.0	4.1	-	nd	-	2.0	6.5
C_16:1_ ω9c	-	-	13.8	-	12.9	-	-	-	-	nd	-	-	-
C_18:2_ ω6,9c	-	13.5	-	-	-	-	-	-	Nd	nd	-	-	-
C_18:1_ ω7c	4.6	2.6	3.3	4.3	2.7	4.4	3.8	4.2	-	nd	-	-	-
C_16:1_ ω7c	-	-	2.4	0.9	3.0	0.8	0.8	1.0	-	nd	-	-	-

^1^ nd, not determined; ni, not indicated. -, fatty acids representing less than 0.7% or not detected were omitted. +, fatty acid present without percentages indication.

## Data Availability

The data presented in this study are available in Appendix A and via accession numbers described in the Section 3.6 of this article.

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
