# Peer review of "Expanding the Bacterial Diversity of the Female Urinary Microbiome: Description of Eight New *Corynebacterium* Species"

_microorganisms, 2023, doi:10.3390/microorganisms11020388_

Round 1
Reviewer 1 Report
The manuscript entitled "Description of Corynebacterium lehmanniae, Corynebacterium 2 meitnerae, Corynebacterium evansiae, Corynebacterium 3 curieae, Corynebacterium macclintockiae, Corynebacterium 4 hessae, Corynebacterium marquesiae, and Corynebacterium 5 yonathiae isolated from the urine of healthy women and female 6 patients with overactive bladder", although a very interesting paper regarding the new found of new species is very poorly presented. The title not summarized what is described in the paper and neither does the abstract. The origin of urine is not very well understood and there is no information regarding the origin of the sample, the clinical information of the women and why only women's urine was used.
The aim of this paper is very general should be more precise.
The discussion and conclusion almost only refer the discovery of new species that is important yes, but the importance of this bacteria and the imapct on health is forgotten and should be develop, or the manuscrip should only focus on the description of new species
In the title, the name of the organisms should be in italic
Abstract: conclusions missing and aim
Introduction: develop the aims of the study
Author Response
1- The title not summarized what is described in the paper and neither does the abstract.
The title was rewritten, according to the suggestion of reviewer 1 and 2 (comment 1). Moreover, we made modifications to the abstract.
2- The origin of urine is not very well understood, and there is no information regarding the origin of the sample, the clinical information of the women and why only women's urine was used.
We included a more detailed description of the origin of the samples, participants, and clinical information in Material and Methods, section 2.1 and 2.2. We included only urine samples from women because they were part of an ongoing Female Urinary Microbiome project conducted by our research group. This information was included in Abstract and Introduction (lines 64-66).
3- The aim of this paper is very general should be more precise.
The aim of the paper was rewritten (lines 67-69).
4- The discussion and conclusion almost only refer the discovery of new species that is important yes, but the importance of this bacteria and the impact on health is forgotten and should be develop, or the manuscript should only focus on the description of new species.
The aim of our study was to describe and characterize eight new Corynebaterium species, which will definitively increase the knowledge of members that compose the female urinary microbiome. The importance and the impact of these new species on health or even disease was not addressed in this study, being difficult to elucidate their role. Nevertheless, we discussed the role of other Corynebacterium species, known to have a protective role in other human body sites (lines 552-559).
5- In the title, the name of the organisms should be in italic.
We changed the title according with suggestions of reviewer 2 (comment 1).
6- Abstract: conclusions missing and aim.
The aim and conclusions of this study were included in the abstract.
7- Introduction: develop the aims of the study
The aim of this study was rewritten according with reviewer 2 (comment 2).
Reviewer 2 Report
Thank you for submitting your work to our Journal.
While the amount and novelty of your work cannot be questioned, I have some comments.
First, your title is long and not appealing for the potential reader. You should consider changing it for something more concise and attractive.
The aim of the study should be more clearly defined, ex: "we aim to identify and describe the bacterial species that compose ..."
In my vision, the whole content of the conclusion part actually belongs to Results.
You should still add some words as conclusions of your work, like : "our study managed to identify and describe ... Further work is required to confirm and validate our results.
I suggest you cite a review paper on overactive bladder and experimental techniques: A review of prospective Clinical Trials for neurogenic bladder: The place of surgery, experimental techniques and devices - doi: 10.5173/ceju.2014.03.art12
The way I see it, your paper requires some repositioning of large parts of its content and stronger statements regarding the aims and conclusions. While describing the methods, I suggest you think about someone trying to exactly reproduce your work and you give him instructions as clear as possible.
I will gladly review a modified version of your paper.
Author Response
1- First, your title is long and not appealing for the potential reader. You should consider changing it for something more concise and attractive.The title was rewritten according to the reviewer's suggestion.
2- The aim of the study should be more clearly defined, ex: "we aim to identify and describe the bacterial species that compose ..."
The aim of the paper was rewritten (lines 67-69).
3- In my vision, the whole content of the conclusion part actually belongs to Results.
We included the description of Corynebacterium sp. nov. in the section of results (3.6. Description of Corynebacterium sp. nov.) according with the reviewer suggestion.
4- You should still add some words as conclusions of your work, like : "our study managed to identify and describe ... Further work is required to confirm and validate our results.
We changed the conclusion.
5- I suggest you cite a review paper on overactive bladder and experimental techniques: A review of prospective Clinical Trials for neurogenic bladder: The place of surgery, experimental techniques and devices - doi: 10.5173/ceju.2014.03.art12
We thank the referee for mentioning this reference. Nevertheless, in our paper surgical and experimental techniques used in the management of OAB are not discussed.
6- The way I see it, your paper requires some repositioning of large parts of its content and stronger statements regarding the aims and conclusions. While describing the methods, I suggest you think about someone trying to exactly reproduce your work and you give him instructions as clear as possible.
As previously stated in comment 4, we repositioned the description of Corynebacterium sp. nov. in the section of results (3.6. Description of Corynebacterium sp. nov.). The aim and conclusions of this study were changed according with the reviewer suggestions. The information concerning participants, sample collection and processing was previously published (https://doi.org/10.1128/spectrum.01308-22, https://doi.org/10.1016/S1569-9056(19)30077-6), yet we included it in lines (72-92). The remaining methodology associated with characterization of Corynebacterium strains was detailed in our previous version. If there are additional doubts, please specify it.
Round 2
Reviewer 1 Report
The authors improved the manuscript and now should be accept in the present form.
Reviewer 2 Report
Thank you for making the changes I suggested.